# Comparative Analysis of Compound Probiotics, Seasonal Variation, and Age on Gut Microbial Composition and Function in Endangered Forest Musk Deer

**DOI:** 10.3390/microorganisms13091991

**Published:** 2025-08-26

**Authors:** Feng Jiang, Pengfei Song, Zhenyuan Cai, Guosheng Wu, Shunfu He, Haifeng Gu, Hongmei Gao, Tongzuo Zhang

**Affiliations:** 1Key Laboratory of Adaptation and Evolution of Plateau Biota, Northwest Institute of Plateau Biology, Chinese Academy of Sciences, Xining 810001, China; jiangfeng@nwipb.cas.cn (F.J.); pfsong@nwipb.cas.cn (P.S.); caizhenyuan@nwipb.cas.cn (Z.C.); guhaifeng@nwipb.cas.cn (H.G.); gaohm@nwipb.cas.cn (H.G.); 2Qinghai Provincial Key Laboratory of Animal Ecological Genomics, Xining 810001, China; 3Xining Wildlife Park, Xining 810016, China; 13997227592@163.com (G.W.); heshunfu_xnzoo@126.com (S.H.)

**Keywords:** endangered wildlife, gut microbiota, compound probiotics, contribution ranking, 16S rRNA sequencing

## Abstract

Due to persistent poaching and habitat fragmentation, wild forest musk deer (*Moschus berezovskii*) in China have sharply declined; although captive breeding helps, frequent gut diseases limit further expansion. This study used high-throughput 16S rRNA sequencing to analyze the effects of age, season variation, and compound probiotics on the gut microbiota of captive individuals. The results demonstrated that compound probiotics exerted a significantly greater influence on gut microbial composition, α-diversity, and functional variation compared to the effects of age or seasonal factors. β-diversity analysis confirmed greater differences between probiotic-treated and control groups than among age or seasonal groups. Microbial community assembly was mainly driven by deterministic processes, with stochastic processes also playing a role in winter. Compound probiotics markedly reshaped dominant bacterial taxa at both phylum and genus levels, with *Acinetobacter* identified as a key biomarker. They also significantly modulated metabolic and phenotypic traits, decreasing functions related to Gram-positive and aerobic bacteria while enhancing those linked to Gram-negative characteristics. Environmental correlation analysis further demonstrated that compound probiotics exerted a stronger influence than both age and seasonal factors. The findings underscore the value of dietary and probiotic strategies for enhancing gut health and resilience in endangered forest musk deer.

## 1. Introduction

The genus *Moschus* belongs to the family Moschidae of the order Artiodactyla and is the only extant genus in this family. Currently, there are seven known species of musk deer worldwide [1,2], which are endemic to Eastern Asia. Musk secreted from the musk gland of adult male musk deer is one of the traditional precious animal medicines in China and is also a natural high-grade fragrance. It has been widely used in China for a long time and has extremely limited resources, but possesses high medicinal and economic value [3]. China is the country with the richest diversity, abundance, distribution area, and musk deer reserves globally [4]. Its musk resources account for over 70% of the global total, and its musk production contributes to over 90% of the global output [1,4]. Over the past 70 years, long-term illegal hunting and habitat fragmentation have caused a sharp decline in wild musk deer populations. China’s musk deer numbers dropped from about three million in the 1950s to fewer than 70,000 by the early 21st century, representing a decline exceeding 97% [5,6]. To curb the sharp decline in the musk deer population, China initiated artificial breeding and reproduction research on musk deer in 1958, which proved to be an effective measure for the translocation conservation of wild musk deer and the sustainable utilization of musk. Among them, forest musk deer (*Moschus berezovskii*) are not only the earliest type to be bred in China, but also the most important group of musk deer artificially bred currently, with a breeding scale far exceeding that of other types. In 2009, the Chinese national survey on key terrestrial wildlife resources reported that the total population of wild musk deer in China was less than 75,000, with approximately 31,800 forest musk deer in the wild [6]. This species has been assessed as “Endangered” (EN) by the IUCN and listed in Appendix II of the CITES convention (Convention on International Trade in Endangered Species of Wild Fauna and Flora). It is also evaluated as “Critically Endangered” (CR) in the China Red List. Compared to their wild counterparts, captive forest musk deer are more susceptible to intestinal diseases caused by gut microbiota imbalance, with a higher incidence rate [7] and an approximately 30% mortality rate [8]. Intestinal diseases are particularly prevalent during winter and autumn seasons, serving as significant constraints on the scale of forest musk deer farming and impeding its long-term development [9,10].

In recent years, the gut microbiome has been referred to as the “second genome” and has become a current research hotspot. Advances in sequencing technologies have facilitated extensive investigations into the composition and diversity of animal gut microbiota, with high-throughput 16S rRNA sequencing and metagenomic approaches being widely applied in diverse aspects of endangered species conservation. This has also provided new research tools for wildlife conservation studies [11,12]. Although there are certain differences between fresh fecal samples and the microbiota within the host’s gut, fecal samples can represent the characteristics of certain host gut microbiota in terms of composition and function [13,14]. Meanwhile, wild populations of endangered species are relatively small, and obtaining relevant experimental samples is difficult. Fecal samples have many advantages, such as being easy to collect, preserve, and non-invasive for wildlife. Therefore, conducting research on endangered species through non-invasive fecal sampling has gradually become one of the optimal research methods in conservation biology [15,16]. Research on the composition, diversity, and functionality of the gut microbiota of endangered animals such as rare mammals and birds, conducted through high-throughput sequencing, plays a crucial role in investigating the genetic mechanisms, survival status, evolutionary adaptation to extreme environments of endangered species, as well as assessing the health status, nutritional, and metabolic levels of wild or captive populations [17,18].

During the long-term co-evolution with the host, the gut microbiota forms a close mutualistic relationship with the host and becomes an integral part of the organism [19]. This complex and dynamic ecosystem, co-constructed with the host microenvironment, not only participates in host metabolism, aiding in food digestion and facilitating nutrient absorption, but also plays indispensable roles in immune regulation, promoting intestinal development, and defense against pathogens [20]. The gut microbiota facilitates food degradation, metabolism of various substances (carbohydrates, proteins, and lipids, etc.), and nutrient absorption in ruminant animals. The forest musk deer belongs to the typical solitary small forest-dwelling ruminants. When ruminants ingest food, it first enters the rumen, where microbial decomposition and metabolism convert the food into small molecules (such as glucose, amino acids, and short-chain fatty acids) that the host can utilize, providing the host with sufficient energy and various nutrients [21]. The gastrointestinal microbiota, as in most animals, is composed of a complex community that includes beneficial probiotic bacteria, opportunistic pathogens, and harmful pathogenic taxa [22]. Opportunistic pathogens, primarily facultative anaerobes, typically constitute non-dominant gut microbial populations and proliferate rapidly under specific conditions such as compromised host resistance or intestinal dysbiosis, exerting pathogenic effects on the host [23]. Pathogenic bacteria like *Clostridia* and *Fusobacteria* are transient residents in the gut and generally do not cause disease under stable intestinal microbiota conditions [24]. During our surveys and sampling in multiple forest musk deer farms, we observed that artificially bred musk deer are also susceptible to infections by *Pseudomonas aeruginosa*, leading to fatalities in several individuals. Additionally, the composition and functionality of the host gut microbiota are influenced by various factors, including diet, season, age, gender, and lifestyle [25,26,27]. Changes in diet composition can rapidly influence the composition and abundance of the host gut microbiota [28]. Similarly, supplementation of food with a compound probiotics mixture also affects the composition and diversity of gut microbiota [29]. Seasonal changes lead to alterations in food composition and availability, resulting in seasonal variations in the structure and function of gut microbiota in various wildlife species [30,31]. Moreover, as the host transitions from infancy to adulthood and eventually aging, alterations occur in the structure and diversity of gut microbial communities, which are essential for maintaining host health [32].

Our previous study indicated that the composition and function of captive musk deer gut microbiota are influenced by season and age [33], as well as by compound probiotics [34]. However, the relative contributions of these typical factors to the composition and function of musk deer gut microbiota remain unclear. Therefore, this study selected captive musk deer of different seasons, ages, and whether fed compound probiotics as research subjects. Non-invasive sampling methods are used to collect fecal samples, and high-throughput sequencing of musk deer fecal samples is performed using the 16S rRNA gene amplification technology on the Illumina Miseq sequencing platform. This study aims to investigate the effects and relative contributions of three key factors—age, season, and the administration of compound probiotics—on the composition, diversity, and functional profiles of the gut microbiota in captive forest musk deer. The study results will provide a scientific basis for effective health management of captive musk deer and are of great significance for evaluating captive breeding environments and future reintroduction plans.

## 2. Materials and Methods

### 2.1. Sample Collection and Processing

Non-invasive sampling was conducted at a forest musk deer farm in Feng County, Shaanxi Province, China, during the summer (mid-July). Fresh fecal samples were collected from adult captive forest musk deer with ages greater than 1.5 years and from juvenile individuals with ages less than 1.5 years. These two groups were referred to as the G1 group (*n* = 10) and the G2 group (*n* = 20), respectively. Similarly, at a forest musk deer farm in Qilian County, Qinghai Province, China, fresh fecal samples were collected from both adult and juvenile captive forest musk deer during two different seasons: summer (mid-July) and winter (late December). The adult and juvenile groups in summer were designated as G3 group (*n* = 20) and G4 group (*n* = 20), respectively. The adult and juvenile groups in winter were designated as G5 group (*n* = 20) and G6 group (*n* = 20), respectively. A compound probiotic was added to the feed at varying levels at the forest musk deer farm in Feng County, while no composite probiotic was added to the feed at the forest musk deer farm in Qilian County. A total of 110 fecal samples from forest musk deer were collected for this study.

Independent and enclosed enclosures were provided at the forest musk deer breeding farms in Qilian County, Qinghai Province, and Feng County, Shaanxi Province. Detailed records were maintained for each captive individual by the farm authorities, including information such as the date of entry or birth, age, sex, and whether compound probiotics were administered. On the evening prior to sampling, each enclosure was thoroughly cleaned, and the animals were confined individually overnight to ensure that fecal samples could be accurately matched to the corresponding individuals. The following morning, fresh fecal samples were collected from each enclosure. To avoid cross-contamination during the sampling process, each sample was collected promptly using disposable polyethylene gloves and placed into sterile self-sealing bags. The samples were labeled and temporarily stored in a vehicle-mounted refrigerator (Meigu, Shenzhen, China) at −20 °C or in dry ice. Upon returning to the laboratory, all samples were transferred to an ultra-low temperature freezer (Meiling, Hefei, China) at −80 °C for subsequent DNA extraction.

### 2.2. Extraction of Musk Deer Fecal DNA, Amplification of 16S rRNA Gene, and Purification

Total DNA from musk deer fecal samples was extracted following the instructions of the E.Z.N.A.^®^ soil DNA kit (Omega Bio-Tek, Norcross, GA, USA). The purity and concentration of the extracted total DNA from the musk deer fecal samples were measured using a NanoDrop 2000 ultra microspectrophotometer (Thermo Fisher Scientific, Waltham, MA, USA), and the OD260/280 and OD260/230 values were recorded. Additionally, 1% agarose gel electrophoresis was performed at 5 V/cm for 20 min to assess the integrity of the total DNA extracted from musk deer feces.

The bacterial 16S rRNA gene V4-V5 variable region in musk deer fecal samples was amplified by PCR using universal primers (515F and 907R). The primer sequences are as follows: 515F: 5′-GTGCCAGCMGCCGCGG-3′; 907R: 5′-CCGTCAATTCMTTTRAGTTT-3′. The PCR reaction system (20 μL) consisted of 5×TransStart FastPfu Buffer 4 μL, 2.5 mM dNTPs 2 μL, forward primer (5 μM) 0.8 μL, reverse primer (5 μM) 0.8 μL, TransStart FastPfu DNA Polymerase 0.4 μL, template DNA 10 ng (with ddH2O as a blank control), and adjusted to 20 μL. Each sample was run in triplicate. The PCR amplification program was as follows: initial denaturation at 95 °C for 3 min; 27 cycles of denaturation at 95 °C for 30 s, annealing at 55 °C for 30 s, and extension at 72 °C for 45 s; final extension at 72 °C for 10 min, and then stored at 4 °C.

Each fecal sample was subjected to PCR amplification in triplicate, and the resulting products were pooled. Three microliters of the PCR product were loaded onto a 2% agarose gel for electrophoresis. After electrophoresis, the gel was visualized using a gel documentation system to observe the PCR amplification products. The PCR products were purified and recovered using the AxyPrep DNA Gel Extraction Kit, and the recovered PCR products were quantified using a microplate fluorometer (Quantus™ Fluorometer) (Promega, Madison, WI, USA).

### 2.3. Construction of Sequencing Libraries and Sequencing on the Illumina-MiSeq Platform

Following the instructions of the library preparation kit (NEXTFLEX Rapid DNA-Seq Kit) (Bioo Scientific, Austin, TX, USA), sequencing libraries were constructed from the purified PCR products. The qualified libraries were subjected to pair-end (PE300) sequencing on the Illumina-MiSeq platform. The library construction and sequencing were primarily undertaken and completed by Majorbio Bio-Pharm Technology Co., Ltd. (Shanghai, China)

### 2.4. Sequencing Data Processing and Bioinformatics Analysis

First, the sequencing data were demultiplexed according to each sample’s unique barcode sequence, allowing a maximum barcode mismatch of 0 and a maximum of 2 mismatched bases in the primer sequence. Sequences that did not meet these criteria were discarded. The sequence direction was then adjusted to obtain the raw reads for each sample. Quality control of the raw reads was performed using Trimmomatic (version 0.39). This process involved removing barcode and primer sequences, trimming bases with a quality score below 20 from the end of each read, and setting a sliding window of 50 bp; if the average quality score within the window was below 20, bases were cut from the start of the window onwards [33]. After this step, further low-quality and mismatched reads were discarded. The high-quality sequences were then merged using FLASH (fast length adjustment of short reads) software based on the overlap between forward and reverse reads, with a minimum overlap length of 10 bp and a maximum mismatch rate of 0.2 in the overlap region. Reads that could not be merged were discarded. The final output was the raw tag sequences for each sample.

The raw tags contained a significant number of chimera sequences, which primarily resulted from incomplete extensions during the PCR process. These chimeras needed to be removed before subsequent analysis. The UCHIME algorithm, in conjunction with a reference database, was used to identify and eliminate the chimera sequences, resulting in clean tags. All clean tags were analyzed using Uparse software (version 7.1, http://drive5.com/uparse/, accessed on 13 July 2025), and singleton sequences without duplicates were removed. The remaining clean tags were considered valid tags, which were then clustered into operational taxonomic units (OTUs) based on 97% sequence similarity. The singleton-free clean tags were mapped to the OTU representative sequence set. The frequency of clean tags with 97% or higher similarity to a given OTU representative sequence was counted as the frequency for that OTU, resulting in the final raw OTU abundance table.

To obtain the most reliable taxonomic annotation information, RESCRIPt (reference sequence annotation and curation pipeline) software [35] was used in conjunction with the SILVA rRNA database (version 138) following the instructions provided by the RESCRIPt authors to construct a local SILVA 138/16S bacterial annotation database. The RDP Classifier (Version 1.22) software was used to perform taxonomic annotation of the OTU representative sequences with a confidence threshold of 0.8, while also excluding entries annotated as chloroplasts or mitochondria. MAFFT software was used to perform multiple sequence alignment of the OTU representative sequences [36], and IQ-TREE was used to construct a phylogenetic tree [37,38]. The nucleotide substitution model was evaluated and selected by ModelFinder [38].

### 2.5. Statistical Analysis

The OTUs were classified taxonomically, and the abundant information corresponding to the annotation results of each OTU in each sample was calculated. The sample sequences were normalized according to the minimum sample sequence number. Relative abundance bar charts at the phylum and genus levels were generated for samples in different groups (using the “stats” package in R software). Venn diagrams were used to depict core and unique phyla and genera among different groups. Clustering heatmaps (using the “pheatmap” package in R software) were employed to compare the similarity and differences in gut microbiota composition between groups. Alpha diversity (α diversity) reflected the diversity of gut microbiota composition. The Sobs index (the observed OTUs), Shannon index, Chao1 index, and PD (phylogenetic diversity) index were selected to analyze the diversity of the gut microbiota composition in forest musk deer. The Sobs and Chao1 indices reflected the richness of gut microbiota, while the Shannon and PD indices reflected the diversity of richness. Qiime software (version 1.9.1) was used to calculate the four α diversity indices. The Wilcoxon rank-sum test was employed to analyze the significance of differences in the four α diversity indices between different groups.

Beta diversity (β diversity) was analyzed to compare the species diversity between different microbial communities and explore the similarities or differences in community composition among different sample groups. PCoA (principal co-ordinates analysis) and NMDS (non-metric multidimensional scaling) were used for β diversity analysis between different groups (packages “vegan” in R software). PCoA is an unconstrained data dimension reduction analysis method used to study the similarities and differences in microbial communities among samples. NMDS is a method that reflects the information of taxa contained in the samples as points in a multidimensional space; the distance between points indicates the degree of difference between different samples, resulting in a spatial positioning plot of the samples. The distances between samples at the OTU level were calculated using 29 similarity distance algorithms. ANOSIM (analysis of similarities) and Adonis (permutational MANOVA) were used to test the differences between groups (packages “vegan”, anosim function) [39], and the R and R^2^ values for the differences were calculated. At the phylum and genus levels, Wilcoxon rank-sum tests were conducted to analyze the differences in gut microbiota of musk deer among different age groups, seasons, and whether they were fed with probiotics. To identify differentially abundant taxa between groups at the genus level, LEfSe software was employed to analyze significant differential taxa among different age and gender groups of musk deer. Linear discriminant analysis (LDA) scores were used to quantify the impact of differential taxa on the differences between groups, with LDA filtering thresholds set at intervals of 0.5 from 0.5 to 5.0, resulting in a total of 10 different thresholds. LDA scores quantitatively assess the enrichment level of specific microbial taxa across groups and their contribution to intergroup differences, with higher scores indicating a more significant role in distinguishing between groups.

Functional prediction was conducted using PICRUSt (Phylogenetic Investigation of Communities by Reconstruction of Unobserved States) [40]. The original OTU representative sequences and original OTU abundance table were used as input data in QIIME2 [41] software. Closed-reference clustering at 97% sequence similarity was performed using the Greengene 13_5 database [42] as a reference (qiime vsearch cluster-features-closed-reference). The resulting data were then uploaded to the online analysis platform PICRUSt for standardization processing and functional prediction. Functional abundance information at multiple levels, such as KEGG Orthology (KO), pathway, and COG functions, was obtained by aligning with the KEGG and EggNOG databases. Wilcoxon rank-sum tests were used to analyze inter-group differences in metabolism-related functions.

## 3. Results

### 3.1. Analysis of Gut Microbiota Composition and α-Diversity in Captive Forest Musk Deer Under Different Influencing Factors

The sequencing of 110 fecal samples from captive forest musk deer yielded raw reads, which were optimized, filtered, and quality-controlled. After 97% similarity clustering, a total of 2912 valid OTUs were identified from the fecal samples, which were classified into 19 phyla, 31 classes, 79 orders, 147 families, and 344 genera.

At the phylum level, Firmicutes, Bacteroidetes, Proteobacteria, and Actinobacteria were dominant phyla (relative abundance > 1%) shared among the gut microbiota of captive forest musk deer, influenced by the addition of compound probiotics, different seasons, and ages. In captive forest musk deer not receiving compound probiotics, Firmicutes and Bacteroidetes were the dominant phyla, together accounting for over 95% of the relative abundance. Conversely, in individuals supplemented with compound probiotics, the relative abundance of Proteobacteria notably increased, emerging as a dominant phylum particularly in the juvenile group (Figure 1a; Appendix A).

Based on the clustering heatmap analysis of the top 50 most relatively abundant and identifiable bacterial genera, it was observed that samples from the Shaanxi Feng County farm (fed with compound probiotics) and the Qinghai Qilian County farm (not fed with compound probiotics) each formed distinct clusters. Additionally, the samples from the Qinghai Qilian County farm, both adult and juvenile, clustered separately according to the summer and winter seasons (Figure 1b). Among the top 50 identifiable bacterial genera, *Christensenellaceae R-7 group*, *UCG-005*, *Acinetobacter*, *Bacteroides*, *Rikenellaceae RC9 gut group*, *Alistipes*, *NK4A214 group*, *Prevotellaceae UCG-004*, *Ruminococcus*, and *Monoglobus* were dominant genera (relative abundance > 1%), all belonging to either the Firmicutes or Bacteroidetes phyla (Figure 1b; Appendix A).

Venn diagram analysis results showed that the six groups shared 18 dominant phyla and 268 bacterial genera. The gut microbiota of the Shaanxi Feng County farm had 1 unique phylum and 18 unique genera, while the gut microbiota of the Qinghai Qilian County farm had no unique phyla and one unique genus (Figure 1c). The ternary phase diagram analysis indicated that the distribution of the top 50 bacterial genera was uneven, with genera under the phylum Proteobacteria showing a tendency to enrich in the gut microbiota of forest musk deer fed with composite probiotics (Figure 1d). The analysis of the proportion of shared OTUs revealed that the proportion of shared OTUs was highest in the age comparison group, followed by the seasonal comparison group, and lowest in the compound probiotics comparison group (Figure 1e; Appendix A). This indicated that the compound probiotics comparison group had the greatest differences, while the age comparison group had the least differences.

For the gut microbial communities, the Sobs index, Shannon index, Chao1 index, and PD index reflected the alpha diversity status of the gut microbiota in different groups of captive musk deer. The study indicated that the α diversity of the gut microbiota decreased in the musk deer fed with compound probiotics. It was higher in winter than in summer, and there was no significant difference between the adult and juvenile groups. Overall, the α diversity differences in the compound probiotics comparison group were higher than those in the seasonal and age comparison groups (Figure 1f).

### 3.2. Analysis of Beta Diversity in Gut Microbial Composition of Musk Deer Under Different Influencing Factors

β diversity reflected the differences and similarities in the gut microbiota composition of musk deer across different groups. ANOSIM and Adonis tests were used to examine the differences in gut microbiota composition among musk deer of different ages. PCoA analysis based on the Bray–Curtis distance algorithm revealed a distinct separation between the two groups fed compound probiotics and the four groups not fed probiotics. Among the four groups without probiotics, the two summer groups and the two winter groups also showed a clear separation. ANOSIM test indicated significant differences in gut microbiota composition among the six different groups of musk deer (R = 0.5837, *p* < 0.05) (Figure 2a). NMDS analysis, based on the Bray–Curtis distance algorithm, showed a stress value of 0.165, less than 0.2, which accurately reflected the true arrangement of the data and the differences in gut microbiota composition among the groups, consistent with the PCoA results (Figure 2b). PLS-DA (Partial least squares discriminant analysis) indicated that the addition of compound probiotics and seasonal variations effectively distinguished the six groups (Figure 2c). Based on the ANOSIM test using 29 different distance algorithms (Appendix A), analysis of the R values for intergroup differences indicated that the degree of difference between groups with and without compound probiotics was significantly higher than that between seasonal and age groups (Figure 2d). Similarly, the Adonis test using 29 different distance algorithms and analysis of the R2 values for intergroup differences revealed that the degree of difference between groups with and without compound probiotics was also significantly higher than that between seasonal and age groups (Figure 2e).

To analyze the assembly processes of gut microbiota in forest musk deer, this study first examined the contribution ratios of different ecological processes in deterministic and stochastic frameworks to the gut microbiota structures across various groups. During the summer, the gut microbiota of both adult and juvenile forest musk deer, regardless of whether composite probiotics were administered, were predominantly influenced by heterogeneous selection (HeS), a deterministic process. Conversely, the gut microbiota of adult and juvenile forest musk deer in winter was co-dominated by heterogeneous selection (HeS) and dispersal limitation (DL), with the former classified as a deterministic process and the latter as a stochastic process (Figure 2f). Subsequently, the study calculated the β nearest taxon index (βNTI) using a null model. For most of the gut microbiota in adult and juvenile forest musk deer during the summer, regardless of probiotic supplementation, the βNTI values were >2, indicating that heterogeneous selection (HeS), a deterministic process, was the dominant force. In contrast, for most of the gut microbiota in adult and juvenile forest musk deer during the winter, the βNTI values spanned both βNTI > 2 and |βNTI| < 2, suggesting that the gut microbiota was co-dominated by heterogeneous selection (HeS) and dispersal limitation (DL), a stochastic process, consistent with the previous findings (Figure 2g). Additionally, this study calculated the normalized stochasticity ratio (NST) index using the null model and analyzed intergroup differences. Unlike the βNTI index, the NST index considers the phylogenetic relationships between microbiota. The results showed that only the gut microbiota of juvenile forest musk deer fed composite probiotics had an NST index greater than 50%, indicating a stochastic assembly process, while the other five groups had NST indices below 50%, indicating deterministic assembly processes (Figure 2h).

The neutral community model (NCM) was used to analyze the assembly mechanisms affecting gut microbiota in different groups. The study indicated that the overall community variance among the six different groups (represented by R^2^, which indicates the overall fit of the neutral community model) was only 33.77%, which was relatively low. This suggests that the community assembly was not close to the neutral model and was more influenced by deterministic processes rather than stochastic processes (Figure 2i). Separate analyses for each group revealed that the R^2^ values for G1, G2, G3, G4, G5, and G6 groups were 45.64%, 45.71%, 45.46%, 43.42%, 53.84%, and 50.25%, respectively. These results indicate that in summer, the gut microbiota community assembly of musk deer populations with and without compound probiotics was more influenced by deterministic processes, while in winter, the community assembly of musk deer populations without compound probiotics was more influenced by stochastic processes (Appendix A). The migration rates (*m* values) at the community level for the six different groups were 0.2228, 0.1109, 0.1156, 0.1205, 0.1686, and 0.1869, respectively. Overall, this indicated that the dispersal of gut microbiota in musk deer was higher in winter than in summer; among adults, the dispersal was higher in those without compound probiotics than in those with probiotics, while the opposite trend was observed in juveniles. In summary, based on the four different analysis methods, the gut microbiota assembly process of most forest musk deer was predominantly deterministic.

### 3.3. Analysis of Intergroup Differences in Dominant Gut Microbiota of Forest Musk Deer Under Different Influencing Factors

At the phylum level, a Wilcoxon rank-sum test was used to analyze the differences in the gut microbiota of forest musk deer under various influencing factors. The results showed that the relative abundance of Firmicutes significantly decreased after the addition of compound probiotics, while Bacteroidetes also exhibited a decreasing trend (Figure 3a). The Firmicutes to Bacteroidetes ratios showed no intergroup differences between the groups with and without composite probiotics (G1–G3, G2–G4) and among the age groups (G1–G2, G3–G4, G5–G6). However, significant intergroup differences in the Firmicutes to Bacteroidetes ratios were observed between the seasonal groups (G3–G5, G4–G6) (Figure 3b). The relative abundances of Proteobacteria and Actinobacteria significantly increased (Figure 3a). There was some seasonal variation in the relative abundances of Firmicutes and Bacteroidetes, whereas the seasonal differences in the relative abundances of Proteobacteria and Actinobacteria were not significant. Additionally, the intergroup differences in dominant phyla between adult and juvenile groups were not significant. Overall, the addition of compound probiotics significantly altered the relative abundance of dominant gut microbiota in forest musk deer, with the impact on dominant phyla being greater than that of seasonal and age factors.

At the genus level, the addition of compound probiotics significantly altered the relative abundances of dominant genera such as *Christensenellaceae R-7 group*, *Acinetobacter*, and *Bacteroides* (Figure 3c). Among the groups with and without compound probiotics, 57.14% (16/28) exhibited significant intergroup differences. Additionally, 50.00% (14/28) showed significant seasonal differences, and 26.19% (11/42) showed significant age-related differences (Figure 3c; Appendix A). Overall, it was evident that the impact of adding compound probiotics on the gut microbiota in forest musk deer was greater than that of seasonal and age factors.

Using LEfSe analysis, the significant differential taxa among different groups of forest musk deer were identified with LDA thresholds set at intervals of 0.5 from 0.5 to 5.0. The study indicated that as the LDA threshold increased, the number of significant differential taxa gradually decreased. At the same LDA threshold, the number of significant differential taxa in the groups with and without compound probiotics was higher than that in the seasonal groups, with the age groups having the lowest number of significant differential taxa (Figure 3d).

The random forest model was used to analyze and identify important biomarkers in the gut microbiota of forest musk deer under different influencing factors. At the OTU level, the two groups fed with composite probiotics (G1 and G2 groups), the two groups not fed with probiotics during summer (G3 and G4 groups), and the two groups not fed with probiotics during winter (G5 and G6 groups) exhibited a distinct triangular pattern, with low similarity among the major groups and high similarity within each subgroup (Figure 3e). The taxa importance ranking analysis for the biomarkers indicated that OTU990, OTU4420, and OTU1004 had the highest scores, with both OTU990 and OTU1004 belonging to the genus *Acinetobacter* (Figure 3f). Similarly, the random forest analysis at the genus level confirmed that *Acinetobacter* was still the highest-ranking biomarker (Figure 3g). This indicated that the genus *Acinetobacter* was a significant biomarker for distinguishing differences under various influencing factors.

### 3.4. Combined Analysis of the Impact of Different Influencing Factors on the Composition and Diversity of the Gut Microbiota in Forest Musk Deer

The length of the environmental factor arrows in the CCA analysis represented the extent of the impact (explained variance) of these factors on the gut microbiota composition. The results indicated that the explanatory value of the seasonal group was slightly higher than that of the compound probiotics group, and both were significantly higher than that of the age group (Figure 4a). The distance-based redundancy (db-RDA) analysis based on the Bray–Curtis distance algorithm showed that the explanatory value of the compound probiotics group was slightly higher than that of the seasonal group, and both were higher than that of the age group (Figure 4b). The Mantel test analysis, which calculated the r values under 30 different distance algorithms, showed that the absolute value of r indicated the magnitude of the intergroup differences (Appendix A). The analysis of the absolute values of r demonstrated that the differences in the compound probiotics group were significantly higher than those in the seasonal and age groups (Figure 4c).

Mantel test results showed that the correlation between the probiotic group and the seasonal group, and the probiotic group and the age group, were 0.46 (*p* < 0.001) and −0.15 (*p* > 0.05), respectively, while the correlation between the seasonal group and the age group was −0.07 (*p* > 0.05), indicating a significant positive correlation between the probiotic group and the seasonal group (Figure 4d). MaAsLin2 analysis retained significant correlation coefficients (*p* < 0.05), revealing that at the OTU and genus levels, the number of correlation coefficients between the probiotic group and taxa was not only higher than those in the seasonal and age groups, but the coefficient values were also significantly greater (Figure 4e). Procrustes analysis based on least squares at the OTU level showed that for the composition of gut microbiota in forest musk deer under different influencing factors, the correlation between the probiotic group and the seasonal group (M^2^ = 0.502, R = 0.706) was greater than that between the probiotic group and the age group (M^2^ = 0.696, R = 0.551) and between the seasonal group and the age group (M^2^ = 0.730, R = 0.520) (Figure 4f).

Furthermore, analyzing the intergroup differences of 19 bacterial phyla revealed that the proportions of significant differences in the compound probiotics, seasonal, and age groups were 84.21% (16/19), 57.89% (11/19), and 21.05% (4/19), respectively (Figure 4g). By performing environmental factor ordination regression analysis on the Sobs, Shannon, and Chao1 α-diversity indices across different groups, the determination coefficient R^2^ values were calculated (Appendix A). A higher R^2^ value indicated a greater explanatory power of the environmental factor on the sample differences along the ordination axis. For each α-diversity index, the R^2^ values for the compound probiotics group were sequentially higher than those for the seasonal and age groups. For β-diversity, PCA, PCoA, and NMDA were selected for analysis, and the results indicated that the R^2^ values for the compound probiotics group were also sequentially higher than those for the seasonal and age groups for each β-diversity analysis method. The intergroup difference analysis of the R^2^ values showed that the R^2^ values for the compound probiotics group were significantly higher than those for the seasonal and age groups (Figure 4h). VPA analysis revealed that the explanatory power of compound probiotics, season, and age on community differences was 31.09% (individual explanatory power 23.82%), 7.17% (individual explanatory power 1.75%), and 3.90% (individual explanatory power 1.54%), respectively. The combined explanatory power of the three factors was 34.35%, leaving a residual explanatory power of 65.65%, indicating that the impact of compound probiotics on the differences in forest musk deer gut microbiota was higher than that of seasonal and age factors (Figure 4i).

### 3.5. Analysis of Functional Differences in the Gut Microbiota of Forest Musk Deer Across Different Influencing Factors

Based on the EggNOG database annotation results, Mann–Whitney U test was performed to analyze the differences in 24 functions (at level 2) between each pair of groups and calculated the *p*-values for the differences. The analysis showed that in the compound probiotics group comparisons, the proportions of significantly different functions (*p* ≤ 0.05) were 25.00% (G1–G3, 6/24) and 66.67% (G2–G4, 16/24). In the seasonal group comparisons, the proportions were 70.83% (G3–G5, 17/24) and 45.83% (G4–G6, 11/24). In the age group comparisons, the proportions of significantly different functions were 33.33% (G1–G2, 8/24), 0.00% (G3–G4, 0/24), and 8.33% (G5–G6, 2/24) (Figure 5a). Moreover, based on the annotation results from the KEGG database, 39 functions were selected at level 2 for analysis. Using the same method, it was found that in the comparisons of the compound probiotics group, the proportions of significantly different functions (*p* ≤ 0.05) were 33.33% (G1–G3, 13/39) and 66.67% (G2–G4, 26/39). In the seasonal group comparisons, the proportions were 64.10% (G3–G5, 25/39) and 23.08% (G4–G6, 9/39). In the age group comparisons, the proportions of significantly different functions were 46.15% (G1–G2, 18/39), 0.00% (G3–G4, 0/39), and 0.00% (G5–G6, 0/39) (Figure 5b).

Based on the functional annotation analysis at level three3 of the EggNOG and KEGG databases, among 4544 COG functions, the proportion of significantly different functions (*p* ≤ 0.05) in the compound probiotics group (G1–G3: 53.04%, G2–G4: 80.66%) was higher than that in the seasonal group (G3–G5: 47.95%, G4–G6: 38.45%) and the age group (G1–G2: 51.83%, G3–G4: 8.41%, G5–G6: 10.78%). Among 281 Module functions, the proportion of significantly different functions in the compound probiotics group (G1–G3: 47.33%, G2–G4: 75.44%) was higher than that in the seasonal group (G3–G5: 47.33%, G4–G6: 32.74%) and the age group (G1–G2: 48.04%, G3–G4: 8.19%, G5–G6: 8.19%). Among 2154 Enzyme functions, the proportion of significantly different functions in the compound probiotics group (G1–G3: 52.23%, G2–G4: 79.11%) was higher than that in the seasonal group (G3–G5: 44.34%, G4–G6: 37.19%) and the age group (G1–G2: 49.81%, G3–G4: 9.24%, G5–G6: 9.98%). Among 5746 KO functions, the proportion of significantly different functions in the compound probiotics group (G1–G3: 56.79%, G2–G4: 82.16%) was higher than that in the seasonal group (G3–G5: 41.91%, G4–G6: 36.01%) and the age group (G1–G2: 53.24%, G3–G4: 8.09%, G5–G6: 9.07%). Among the 279 functions at level 3 of the KEGG database, the proportion of significantly different functions in the compound probiotics group (G1–G3: 44.09%, G2–G4: 73.12%) was higher than that in the seasonal group (G3–G5: 47.67%, G4–G6: 32.62%) and the age group (G1–G2: 50.54%, G3–G4: 2.15%, G5–G6: 9.68%). Overall, the differences in various functions in the compound probiotics group were higher than those in the seasonal and age groups (Figure 5c).

Analysis of the trends in changes in eight metabolism-related functions selected from the EggNOG database revealed that in the group administered with compound probiotics, some metabolic functions in the gut microbiota of the musk deer were significantly lower compared to those in the group not administered with compound probiotics. Additionally, in the winter group, the metabolic functions in the gut microbiota of musk deer were significantly higher than those in the summer group. There were relatively fewer diverse functions in the age group. Overall, the proportion of significantly different functions in the compound probiotics group and the seasonal group was higher than that in the age group (Figure 5d). Similarly, analysis of the trends in changes of 11 metabolism-related functions selected from the KEGG database showed similar trends to those annotated in the EggNOG database. Overall, the proportion of significantly different functions in the compound probiotics group was slightly higher than that in the seasonal group, and both were much higher than that in the age group (Figure 5e).

Based on the BugBase database, the phenotypic functions of gut microbiota in forest musk deer under different influencing factors were predicted and intergroup differences analyzed. The results indicated that the phenotypic functions of gut microbiota across different groups included nine major categories, with significant intergroup differences overall (Figure 5f). Generally, the relative abundances of phenotypic types such as Gram-positive, aerobic, Gram-negative, and those containing mobile elements were relatively high. Notably, compared to the groups not fed with composite probiotics, the phenotypic functions of Gram-positive, aerobic, and mobile elements in the gut microbiota of forest musk deer fed with composite probiotics showed a significant decreasing trend, while Gram-negative phenotypic functions exhibited a significant increasing trend (Figure 5g).

## 4. Discussion

The sharp decline in the number of endangered species is one of the urgent challenges facing global biodiversity, with wildlife populations around the world facing severe threats. In order to more effectively protect these rare species, metagenomics and gut microbiota research, as frontier fields, have been widely applied in the conservation of various types of endangered species, including ungulates [43,44], carnivores [45,46], primates [47,48], and birds [49,50]. These studies provide important reference for the conservation and management of endangered species, aiding in the formulation of more scientific conservation strategies and improvements in captive conditions, thereby promoting the reproduction and survival of endangered species. Over the past 70 years, due to long-term hunting pressure and habitat fragmentation, the population size and density of musk deer in China have sharply declined, leading to their classification as a globally endangered species. To alleviate the pressure on wild populations from hunting and to sustainably utilize natural musk resources, China has initiated artificial breeding programs for musk deer, with forest musk deer being the most extensively farmed species in the family. Therefore, this study utilized high-throughput sequencing technology targeting the 16S rRNA gene to investigate the gut microbiota of forest musk deer, an endangered species globally.

The results indicate that the gut microbiota, evolving alongside hosts, forms a complex microbial ecosystem with the gastrointestinal tract of animals, playing an irreplaceable role in food degradation, various metabolic processes, and nutrient absorption in ruminant animals. The composition, structure, and diversity of gut microbiota are simultaneously influenced by various factors, such as different environmental conditions. Our previous studies have pointed out that the dominant bacterial phyla in the gut in captive forest musk deer show seasonal fluctuations, with higher alpha diversity and major metabolism-related functions during the cold season compared to the warm season, indicating that seasonal changes affect the composition, diversity, and metabolism-related functions of gut microbiota in captive forest musk deer [33]. Furthermore, we have also noted that the addition of a composite probiotic to the diet significantly alters the types and relative abundances of dominant bacterial taxa in the gut microbiota of captive forest musk deer, leading to significant changes in alpha diversity, metabolism, and disease-related functions [34]. Additionally, other studies have indicated that the composition, diversity, and functions of gut microbiota change with host age [51,52]. From infancy to adulthood, the gut microbial community of the host gradually stabilizes. Evidence suggests that the composition of gut microbiota was established and gradually perfected in the early stages of host life [53,54]. The composition of rumen microbiota in Tibetan sheep varies significantly with age, with the relative abundance of Bacteroidetes increasing significantly from 18.9% (D0) to 53.9% (D360), while the Firmicutes decreased significantly from 40.8% to 5.9%, and *Prevotella_1* became the dominant taxon after the seventh day [55]. Higher α-diversity indicates a more complex and stable composition of gut microbiota, with stronger resistance to external disturbances and greater adaptability to different environments [56]. It can be observed that for domesticated animals, under the same dietary composition and environmental conditions, age may be an important factor leading to differences in the composition and diversity of host gut microbiota [57]. Therefore, age, season, and the addition of probiotics to the diet are common factors affecting captive endangered species. Which factor contributes more to changes in the gut microbiota composition and function of forest musk deer? This requires further investigation. Therefore, this study primarily focuses on discussing the contributions and rankings of these three influencing factors.

This study indicates that the addition of compound probiotics to the diet significantly alters the composition, shared operational taxonomic units (OTUs), dominant bacteria, and alpha diversity of gut microbiota in captive forest musk deer. The degree of its impact is sequentially higher than that of seasonal factors and age factors. Beta diversity and metabolic function analysis reveal that the inter-group differences in whether compound probiotics are added to the diet are slightly higher than those of the seasonal group, both of which are significantly higher than the age group. A previous study has pointed out that dietary composition is one of the variable factors that can rapidly influence the composition and abundance of gut microbiota [58]. Similarly, adding compound probiotics to the diet can also alter the composition of the host gut microbiota. Probiotics play an important role in controlling various intestinal diseases and promoting overall health by regulating the composition of the host gut microbiota [28]. Studies have shown that compound probiotics significantly improve the production performance, intestinal immunity, and intestinal barrier function of broilers, especially when adding 10 g/kg compound probiotics. By influencing the community structure and function of cecal microbiota, compound probiotics can enhance intestinal barrier and immune function, thereby improving the growth performance of broilers [59]. Probiotics can also increase feed utilization and disease resistance in dairy cows, increase milk production, and improve the diversity of intestinal microbiota in cows [60]. Additionally, spraying compound probiotic fermented liquid in the feeding environment is also beneficial to host health. Studies have shown that spraying compound probiotic fermented liquid significantly alters the microbial composition of farrowing rooms, improves the daily weight gain and weaning weight of piglets, changes the structure and diversity of piglet gut microbiota, and significantly improves the growth performance and immunity of piglets. These beneficial effects are closely related to changes in gut microbiota and host metabolism [61]. This study suggests that the addition of compound probiotics to the diet has the greatest relative and highest contribution to the dominant bacteria, alpha diversity, and function of gut microbiota in captive forest musk deer, which was inseparable from improving the health status of captive forest musk deer.

Different seasons also influence the structure, diversity, and function of mammalian gut microbiota, leading to the co-evolution of gut microbiota and hosts. This seasonal fluctuation is crucial for the health, food digestion, adaptation to extreme environments, and physiological status of hosts and often coincides with changes in food composition and climate. Studies have found that seasonal nutritional changes, such as the relative scarcity of bamboo leaves and protein-rich sprouts, significantly affect the composition and function of gut microbiota in wild giant pandas (*Ailuropoda melanoleuca*) [62]. During hibernation, the abundance of bacterial taxa capable of degrading mucin polysaccharides increases in the gut microbiota of Chinese alligators (*Alligator sinensis*), such as Bacteroides from the genus Bacteroides, along with an increase in the abundance of mucin oligosaccharide-degrading enzymes and carbohydrate-active enzymes. Conversely, during the active period, the abundance of bacteria involved in protein hydrolysis and amino acid fermentation increases in the gut microbiota of Chinese alligators, such as Fusobacteria from the genus Fusobacteria, to enhance food digestion efficiency [63]. A study on the seasonal differences in the gut microbiota of 33 large herbivorous mammals in East African semi-arid grasslands indicated that food diversity had no significant effect on the diversity of gut microbiota. However, food composition was significantly associated with the composition of gut microbiota, and seasonal differences in food composition could explain 25% of the seasonal differences in microbial composition [64]. This study showed that although the explanatory power of whether compound probiotics are added to the diet for the gut microbiota of captive forest musk deer was much higher than that of seasonal factors and age factors, the contribution of seasonal factors to the beta diversity and metabolic function of musk deer gut microbiota was slightly lower than that of adding compound probiotics to the diet. In summary, the explanatory power of whether compound probiotics were added to the diet for the gut microbiota of captive forest musk deer, dominant bacteria, and alpha diversity was much higher than that of seasonal factors and age factors. However, the impact of whether compound probiotics were added to the diet on the beta diversity and metabolic function of musk deer gut microbiota was only slightly higher than that of seasonal factors. Therefore, the three influencing factors did not completely coincide in terms of the composition, diversity, and function of the gut microbiota of captive forest musk deer, but each had its own emphasis.

Given the limitations of the current cross-sectional design, future studies will aim to include longitudinal data to capture temporal dynamics in the forest musk deer gut microbiota. This approach would provide a deeper understanding of how age, seasonality, and feed additive use influence microbial communities over time, and help to clarify potential causal links with host health outcomes. By systematically collecting intestinal samples and health monitoring data, more detailed models can be established to explore associations between specific microbial community components and various health indicators. It is important to note that the compound probiotic group was limited to a single site, and environmental or management factors inherent to the farm location may have contributed to the observed differences in gut microbiota. This represents a potential confounding factor. Future studies should apply probiotic treatments across multiple standardized sites to better isolate probiotic effects from site-specific variables. Furthermore, the long-term effects of feed additives on the diversity and function of the forest musk deer gut microbiota, especially in captive environments, warrant further investigation. Long-term monitoring and experimentation could provide more robust evidence on how feed additives influence the physiological adaptability of forest musk deer. Based on these findings, management strategies for captive populations may be refined, such as adjusting feed formulations and probiotic additive ratios to support the stability and diversity of gut microbiota. Seasonal management should also be optimized, particularly during winter, by providing appropriate feed and environmental conditions to maintain microbial balance. Finally, incorporating metagenomic, metabolomic, or transcriptomic approaches in future studies would allow direct validation of predicted microbial functions and enhance the robustness of functional interpretations. Overall, while this study highlights important patterns in the gut microbiota of captive forest musk deer, the conclusions regarding health benefits should be interpreted cautiously until validated by additional multi-site, longitudinal, and functionally integrative studies.

## 5. Conclusions

This study conducted a detailed analysis of the gut microbial communities of captive forest musk deer under various influencing factors. The results indicated that the addition of composite probiotics, along with factors such as season and age, significantly affected the composition and diversity of the gut microbiota. At the phylum level, Firmicutes, Bacteroidetes, Proteobacteria, and Actinobacteria emerged as the predominant phyla in the gut of captive forest musk deer. Notably, Firmicutes and Bacteroidetes dominated deer without probiotics, while the relative abundance of Proteobacteria increased significantly following probiotic supplementation, particularly in the juvenile group. Additionally, significant differences in microbial composition were observed between summer and winter, while age had a lesser impact. In terms of α-diversity, the addition of composite probiotics led to a decrease in gut microbiota diversity, with winter exhibiting higher diversity than summer, and differences among age groups were not significant. β-diversity analysis, along with Wilcoxon rank-sum tests, LEfSe analysis, and random forest modeling, collectively revealed significant differences in gut microbiota among different groups, particularly between the probiotic and non-probiotic groups. The influence of composite probiotics on gut microbiota surpasses that of season and age. Investigating the community assembly mechanisms of the gut microbiota revealed that the gut microbiota of forest musk deer was primarily driven by deterministic processes, especially in summer, while populations without probiotics were more susceptible to stochastic processes in winter. Overall, the addition of composite probiotics significantly altered the composition and structure of the gut microbiota in forest musk deer, particularly increasing the relative abundance of Proteobacteria, which became an important biomarker for distinguishing between groups. Functional annotation analysis indicated that the impact of composite probiotics on functional differences in the gut microbiota was significantly greater than that of season and age, particularly in metabolic and phenotypic functions, where the probiotic group showed a significant reduction in Gram-positive, aerobic, and mobile element-containing phenotypic functions, alongside a marked increase in Gram-negative phenotypic functions. In conclusion, the addition of composite probiotics played a crucial role in modulating the gut microbiota of captive forest musk deer, with effects that exceeded those of season and age. While direct health outcomes were not assessed in this study, these findings provide a scientific basis for optimizing husbandry management practices and lay the groundwork for future investigations into the potential health benefits of probiotic supplementation.

## Figures and Tables

**Figure 1 microorganisms-13-01991-f001:**
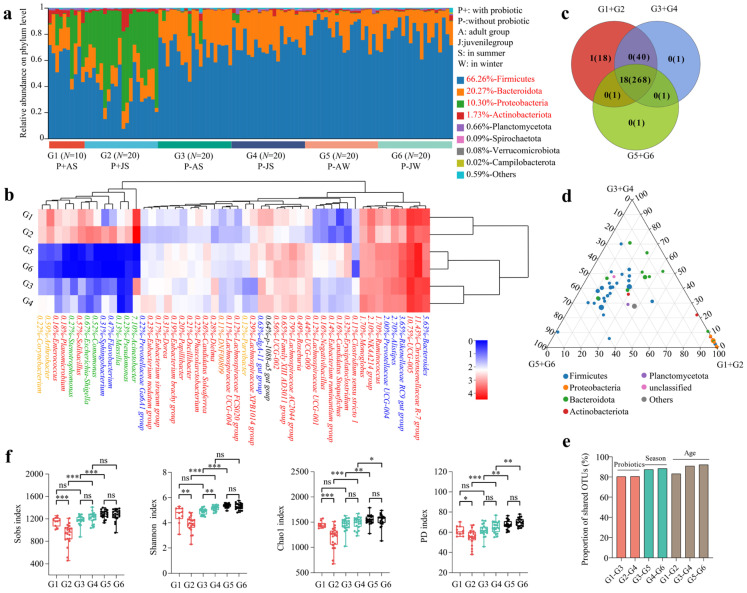
Analysis of gut microbial composition and diversity in musk deer under different influencing factors. (**a**) Analysis of the relative abundance of bacterial composition at phylum level in different groups. Red font indicated dominant phyla. (**b**) Cluster heatmap analysis of the top 50 identifiable bacterial genera in different groups. The bacterial genera marked in red, blue, green, orange, and black in the figure were classified under the phyla Firmicutes (*n* = 31, 62%), Bacteroidota (*n* = 8, 16%), Proteobacteria (*n* = 6, 12%), Actinobacteriota (*n* = 4, 8%), and Planctomycetota (*n* = 1, 2%), respectively. (**c**) Venn diagram analysis of shared and unique bacterial phyla and genera under different factors. (**d**) Ternary plot representing the composition and distribution of the top 50 bacterial genera across different groups. (**e**) Proportion analysis of shared OTUs in different groups. (**f**) Analysis of alpha diversity differences in different groups with four diversity indices.*, *p* < 0.05; **, *p* < 0.01; ***, *p* < 0.001.

**Figure 2 microorganisms-13-01991-f002:**
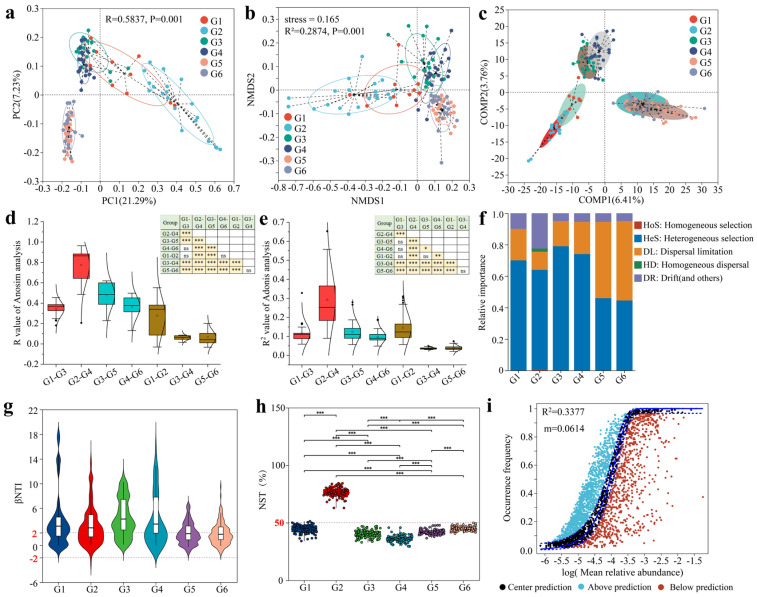
Analysis of β-diversity in the gut microbiota composition of forest musk deer under different influencing factors. (**a**,**b**) PCoA analysis: (**a**) NMDS analysis (**b**) based on Bray–Curtis distance algorithm. (**c**) PLS–DA analysis based on OTU level. (**d**,**e**) Intergroup differentiation analysis of R values from ANOSIM tests (**d**) and R^2^ values from Adonis tests (**e**) using 29 different distance algorithms. (**f**) Relative importance of different ecological processes of gut microbiota in forest musk deer across different groups based on a null model. (**g**) Beta nearest taxon index (βNTI) calculated using the null model and intergroup difference analysis across different groups. (**h**) Normalized stochasticity ratio (NST) analysis based on the null model. (**i**) Analysis of the gut microbiota assembly process based on the neutral community model (NCM). *, *p* < 0.05; **, *p* < 0.01; ***, *p* < 0.001; ns, not significant.

**Figure 3 microorganisms-13-01991-f003:**
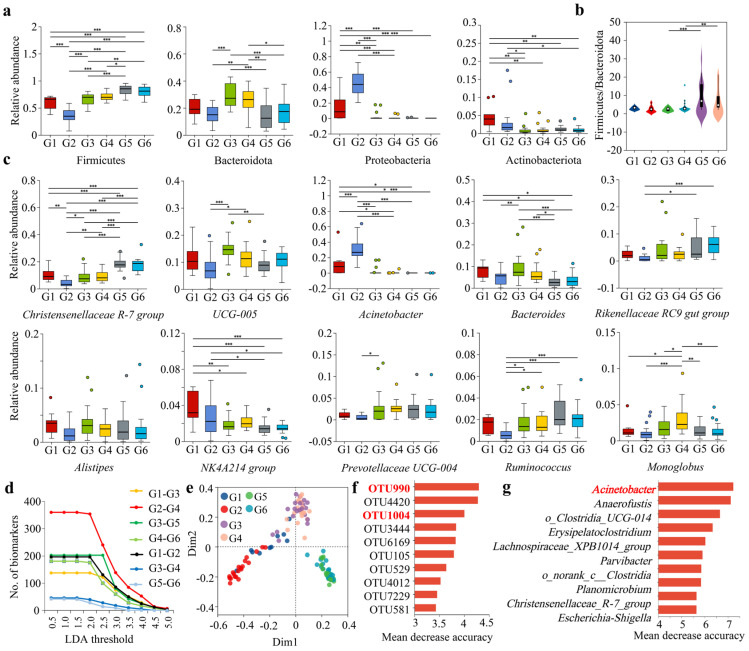
Analysis of differences in dominant gut microbiota and of forest musk deer among different influencing factors. (**a**–**c**) Intergroup differences in dominant bacterial phyla: (**a**) the Firmicutes to Bacteroidetes ratio (**b**) and dominant bacterial genera (**c**) of the gut microbiota in forest musk deer under different influencing factors. (**d**) LEfSe analysis of significant differential taxa among different groups. (**e**) Random forest analysis of intergroup and intragroup similarity under different influencing factors at the OTU level. (**f**,**g**) Ranked importance of indicative gut microbial taxa in different groups at the OTU level (**f**) and genus level (**g**) based on random forest analysis.*, *p* < 0.05; **, *p* < 0.01; ***, *p* < 0.001.

**Figure 4 microorganisms-13-01991-f004:**
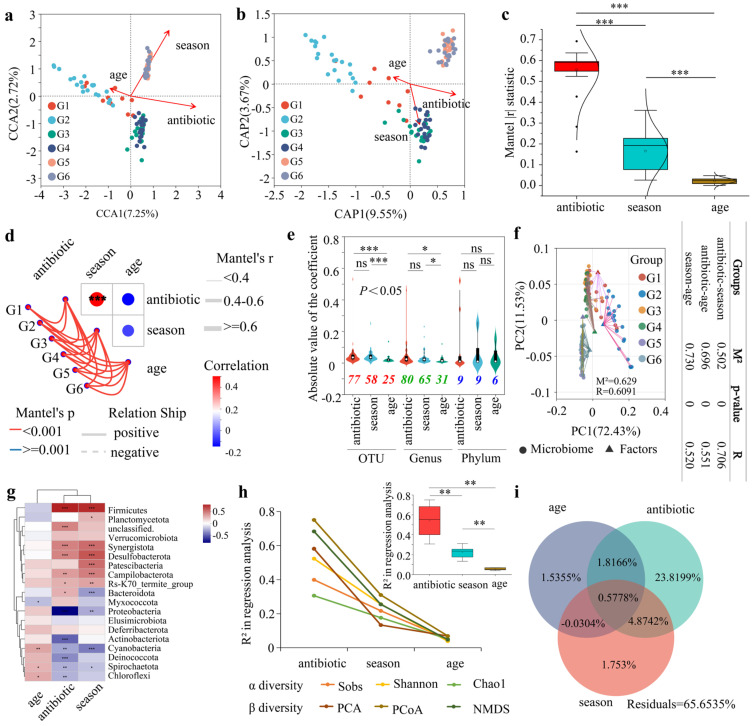
Analysis of the impact of different influencing factors on the composition and diversity of the gut microbiota of forest musk deer. (**a**,**b**) Environmental factor correlation analysis based on CCA analysis (**a**) and db–RDA analysis (**b**) at the OTU level. (**c**) Intergroup difference analysis of the absolute values of *r* from the Mantel test. (**d**) Mantel–test network heatmap of gut microbiota in different groups at the OTU level. (**e**) MaAsLin2 analysis of correlation coefficients between environmental variables and taxa in the gut microbiota under different influencing factors at various taxonomic levels. The red, green, and blue numbers represented the number of correlation coefficients (*p* < 0.05) at the OTU, genus, and phylum levels, respectively. (**f**) Procrustes analysis based on least squares at the OTU level, examining the overall correlation between gut microbial communities and environmental factor. (**g**) Heatmap analysis of the correlation of different influencing factors at the phylum level. (**h**) Environmental factor ordination regression analysis of different diversity indices. (**i**) VPA–based assessment of the explanatory power of different environmental factors on the differences in the gut microbiota.*, *p* < 0.05; **, *p* < 0.01; ***, *p* < 0.001; ns, not significant.

**Figure 5 microorganisms-13-01991-f005:**
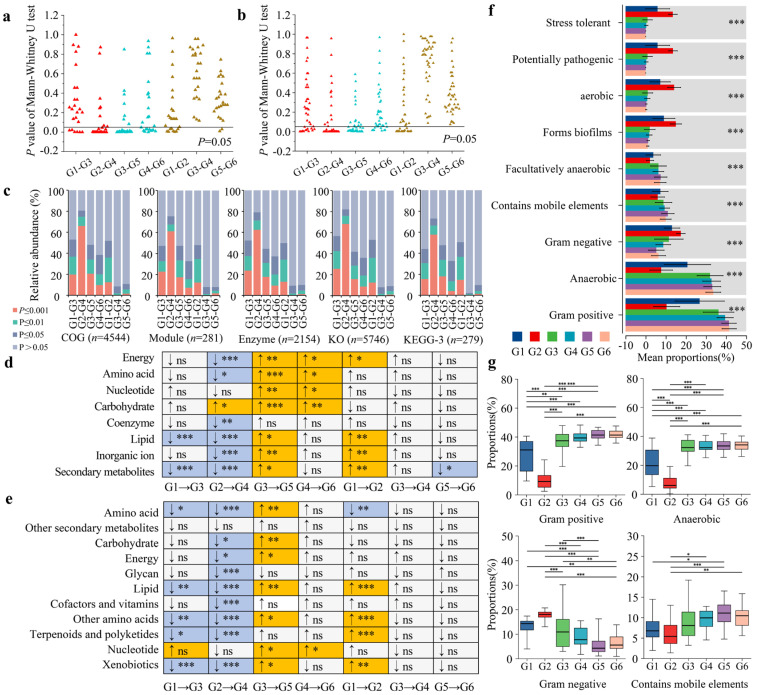
Analysis of the differences in gut microbiota functionality among different influencing factor groups. (**a**,**b**) Statistical analysis of intergroup differences in *p* values of functional annotations at level 2 based on the EggNOG (**a**) and KEGG (**b**) databases. (**c**) Statistical analysis of intergroup differences in *p* values of multi-category functions at level 3 based on annotations from both the EggNOG and KEGG databases. (**d**,**e**) Trend analysis of metabolic function changes based on the EggNOG (**d**) and KEGG (**e**) databases under different influencing factors. (**f**) Relative abundance and overall intergroup difference analysis of phenotypic functions in gut microbiota under different influencing factors based on the BugBase database. (**g**) Trends and intergroup differences in the main phenotypic functions of gut microbiota in different groups.*, *p* < 0.05; **, *p* < 0.01; ***, *p* < 0.001; ns, not significant.

## Data Availability

The datasets of 80 samples generated for this study can be found in the NCBI Sequence Read Archive under BioProject PRJNA725631 with the accession number SUB9547782 (https://dataview.ncbi.nlm.nih.gov/object/PRJNA725631?reviewer=e2rcdic27nir1a7qhp8unlk9s6, accessed on 13 July 2025).

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
