# Peer review of "Comparative Analysis of Compound Probiotics, Seasonal Variation, and Age on Gut Microbial Composition and Function in Endangered Forest Musk Deer"

_microorganisms, 2025, doi:10.3390/microorganisms13091991_

Round 1
Reviewer 1 Report
Comments and Suggestions for Authors
The title is confusing, are they only talking about the impact of probiotics? Don't understand how they relate to the seasonal and age component in deer. Rewrite to be clearer and more concrete
Similarly, the abstract is confusing. They talk about the reduction of deer in the wild. However, is the management given to deer in captivity, with the aim of recovering the population not adequate, it seems that the composition of the probiotics is detrimental to the health of these deer. It is not clear, please rewrite completely.
Please do not use keywords that are already considered in the title.
It is understood that this type of deer is at risk of extinction, however, the subject matter of the manuscript, in the title, focuses on the effect of probiotics on the intestinal health of the animals, and this part, although addressed very limitedly in the introduction, is complicated to understand. Also, the introduction is exaggeratedly long, it seems like a literature review of various aspects of biochemistry, behavior, gut health, etc. This is confusing, so the document is about the conservation status of this species? No state of the art is reflected in the introduction, please adjust accordingly.
The objectives of the study are also several and this makes it difficult to write clearly and concretely. They should be adjusted and include only what is most relevant, or what really helps to understand what the final objective of the manuscript is.
They do not mention any characteristics of the sampled population, characteristics of the place where they were found, age (they only mention adults and juveniles), sex, etc. They immediately address the collection of samples. Add information on the sampled population.
The sample collection periods mention only mid-July and late-December, what was the duration of the sample collection period (in days or weeks)?
It is very complicated to understand the experimental design or whatever the authors are trying to address. They make subdivisions of the groups between “ages” and between “seasons”. It is extremely complex to understand what they mean.
They added a probiotic compound to the captive population for what purpose, why not just compare one against the other and evaluate ages and seasons. They already had the samples of animals in captivity and in the wild, why include a variable that is clearly going to have effects on the intestinal health of the captive population. An apology to the authors, but it is not clear what they mean.
They say they kept detailed individual records of the deer, but what were these records? They don't mention any in the methodology.
Were the animals in separate pens? How did they obtain the fecal samples? There is a complete lack of methodological information that seriously compromises the interpretation and results obtained. This is unacceptable for this type of study.
The methodological description of fecal DNA extraction, sequence library construction, data sequencing and bioinformatics analysis are well stated. There are no major comments in this regard.
In general, the results and discussion are well developed. There are no major comments, however, I emphasize once again that the description of the state of the art and the methodological part describing the populations studied lacks an adequate description that allows to visualize in an integral way the problem that the authors want to address.
Author Response
Comment 1: The title is confusing, are they only talking about the impact of probiotics? Don't understand how they relate to the seasonal and age component in deer. Rewrite to be clearer and more concrete
Response 1: Thank you for the valuable suggestion. We agree with the reviewer’s comment and have revised the manuscript title accordingly. The revised title emphasizes the comparative study of the effects of compound probiotics, season, and age on the composition and function of the gut microbiota in forest musk deer.
Comment 2: Similarly, the abstract is confusing. They talk about the reduction of deer in the wild. However, is the management given to deer in captivity, with the aim of recovering the population not adequate, it seems that the composition of the probiotics is detrimental to the health of these deer. It is not clear, please rewrite completely.
Response 2: Thank you for your valuable comments. In accordance with your suggestions, we have thoroughly revised and rewritten the abstract to improve its clarity, coherence, and overall quality.
Comment 3: Please do not use keywords that are already considered in the title.
Response 3: Thank you for your valuable comments. In accordance with the reviewer’s suggestion, we have thoroughly revised the keywords section of the manuscript. The revised keywords do not repeat the terms used in the title.
Comment 4: It is understood that this type of deer is at risk of extinction, however, the subject matter of the manuscript, in the title, focuses on the effect of probiotics on the intestinal health of the animals, and this part, although addressed very limitedly in the introduction, is complicated to understand. Also, the introduction is exaggeratedly long, it seems like a literature review of various aspects of biochemistry, behavior, gut health, etc. This is confusing, so the document is about the conservation status of this species? No state of the art is reflected in the introduction, please adjust accordingly.
Response 4: Thank you for the valuable comments. We agree with the reviewer’s suggestions and have accordingly revised the manuscript title to better reflect the study focus. The introduction section was then systematically rewritten to align with the revised title. Specifically, the introduction first presents the taxonomic status and biological characteristics of forest musk deer, along with the economic and medicinal value of musk, highlighting the sharp decline of wild populations due to poaching and habitat fragmentation and emphasizing the importance of captive breeding. Next, it addresses the limitations posed by gastrointestinal diseases in captive musk deer, introduces the background and significance of gut microbiota research, and reviews the relationships between gut microbiota and host health, including influences from diet, season, and age. Finally, the introduction clearly states the objectives of this study, which are to evaluate the impacts and relative contributions of compound probiotics, seasonality, and age on the gut microbiota of captive forest musk deer. Additionally, we have refined and condensed the introduction to improve clarity and precision. Regarding conservation techniques for the endangered forest musk deer, our research primarily focuses on leveraging conservation metagenomics approaches.
Comment 5: The objectives of the study are also several and this makes it difficult to write clearly and concretely. They should be adjusted and include only what is most relevant, or what really helps to understand what the final objective of the manuscript is.
Response 5: Thank you for your valuable comments. I have revised the statement of our research objectives in the last paragraph of the introduction. Based on the main focus of our study, I have simplified and clarified the objectives accordingly.
Comment 6: They do not mention any characteristics of the sampled population, characteristics of the place where they were found, age (they only mention adults and juveniles), sex, etc. They immediately address the collection of samples. Add information on the sampled population.
Response 6: Thank you for your valuable comments. During the sample collection process, we recorded information on each individual’s enclosure, age, and sex. This information has been added to the Methods section under “2.1 Sample Collection and Processing.” We also supplemented the age classification criteria for adult and juvenile forest musk deer. Regarding sex as a factor, our previous study showed no significant differences in gut microbiota composition and function between male and female forest musk deer, which has been published. Therefore, sex was not considered as a factor in this study.
Comment 7: The sample collection periods mention only mid-July and late-December, what was the duration of the sample collection period (in days or weeks)?
Response 7: Thank you for the valuable comments. We collected samples from various forest musk deer breeding farms, and the sampling process was generally completed within 1-2 days. Therefore, the samples were collected on a daily basis.
Comment 8: It is very complicated to understand the experimental design or whatever the authors are trying to address. They make subdivisions of the groups between “ages” and between “seasons”. It is extremely complex to understand what they mean.
Response 8: Thank you for your valuable suggestion. The primary objective of our study was to investigate the extent and relative contribution of three distinct influencing factors—age, season, and the use of compound probiotics—on the composition, diversity, and function of the gut microbiota in captive forest musk deer. Specifically, for the factor of age, we conducted a comparative analysis between adult and juvenile individuals. For the seasonal factor, we compared the gut microbiota between two representative periods: summer and winter. Lastly, regarding the use of compound probiotics, we analyzed the differences in gut microbiota between individuals raised with and without compound probiotic supplementation. In response to the reviewer’s previous comment, we have revised the final paragraph of the introduction to clarify and simplify the objectives of our study for better understanding by both reviewers and readers.
Comment 9: They added a probiotic compound to the captive population for what purpose, why not just compare one against the other and evaluate ages and seasons. They already had the samples of animals in captivity and in the wild, why include a variable that is clearly going to have effects on the intestinal health of the captive population. An apology to the authors, but it is not clear what they mean.
Response 9: Thank you for your valuable comments. Firstly, compound probiotics are a type of biological feed additive commonly used in animal husbandry and play a role in the growth and development of captive forest musk deer. As you correctly noted, this study focused on comparing the gut microbiota differences between groups fed with compound probiotics and those not fed with them. During this comparison, we controlled for seasonal effects by selecting samples from the same season and for age effects by selecting individuals at the same age stage, thereby isolating the impact of compound probiotics on the gut microbiota of captive forest musk deer.
Comment 10: They say they kept detailed individual records of the deer, but what were these records? They don't mention any in the methodology.
Response 10: Thank you for the valuable comments. During the sample collection process, information on each forest musk deer’s enclosure, age, and sex was recorded. This information was added to the second paragraph of section 2.1 Sample Collection and Processing.
Comment 11: Were the animals in separate pens? How did they obtain the fecal samples? There is a complete lack of methodological information that seriously compromises the interpretation and results obtained. This is unacceptable for this type of study.
Response 11: Thank you for the valuable comments. Firstly, as the reviewer correctly pointed out, the captive forest musk deer were housed in an enclosed breeding facility, with each individual kept in a separate enclosure. On the night prior to sample collection, all enclosures were thoroughly cleaned, and the animals were kept in their respective enclosures overnight. This procedure ensured that the fecal samples collected corresponded accurately to each individual. Fresh fecal samples were then collected from each enclosure early the next morning. Secondly, we have revised the wording in the Methods section to clarify and supplement the description of the sample collection procedures.
Comment 12: The methodological description of fecal DNA extraction, sequence library construction, data sequencing and bioinformatics analysis are well stated. There are no major comments in this regard.
Response 12: Thank you very much for the valuable comments and for the positive recognition of the methodology used in our study. Thank you.
Comment 13: In general, the results and discussion are well developed. There are no major comments, however, I emphasize once again that the description of the state of the art and the methodological part describing the populations studied lacks an adequate description that allows to visualize in an integral way the problem that the authors want to address.
Response 13: Thank you very much for the valuable comments and for your recognition of the results and discussion sections of our study. In response to your suggestions, we have revised the language and added further details to the Methods section, particularly in the revised “2.1 Sample Collection and Processing” subsection. We sincerely appreciate your insightful feedback and constructive suggestions for improving our work.
Thank you very much for your affirmation of this manuscript and valuable comments again.
Reviewer 2 Report
Comments and Suggestions for Authors
Reviewer Comments
This manuscript presents a well-designed and data-rich study investigating the effects of compound probiotic supplementation on the gut microbiota composition and function of captive forest musk deer, in comparison to seasonal and age-related influences. The research is timely and highly relevant to conservation physiology and animal health, offering valuable insights into microbiome-based interventions for improving the management of endangered species in captivity. The study is scientifically robust, employing high throughput 16S rRNA sequencing, a broad range of diversity and community structure analyses, and functional predictions through multiple databases. However, the confounding of treatment with geographic location, the reliance on predictive rather than empirical functional data, and the lack of longitudinal tracking limit the generalizability and mechanistic depth of the findings. Future studies incorporating metagenomics, metabolomics, and multi-site probiotic interventions are warranted. Minor revisions addressing these limitations would enhance the manuscript’s scientific rigor and conservation impact.
Limitations.
Lack of Longitudinal Data: The study is cross-sectional, capturing snapshots rather than dynamics. This limits conclusions on causality or lasting effects of probiotics.
- Probiotics were only administered in one location (Shaanxi): Thus, effects attributed to probiotics may also reflect environmental or management differences between farms.
- The reliance on PICRUSt-based predictions for functional profiling lacks direct evidence. Metabolomic or transcriptomic data would have strengthened conclusions.
- Although probiotics were the intervention, other dietary components were not strictly detailed, which may influence microbial outcomes. Provide table showing the feed composition of the other groups
- Suggestion for Improvement
- Use shotgun metagenomics or metabolomics in future work to validate predicted functional changes.
- Consider a longitudinal design to monitor the lasting effects of probiotics and seasonal adaptation.
- Implement multi-site probiotic treatments to separate location from treatment effects.
- Clarify and standardize non-probiotic dietary components across groups.
- Please check line 251….To identify differential abundant taxa is safe rather than species…Replace species there with taxa, and do that throughout the paper
Author Response
Comment 1: This manuscript presents a well-designed and data-rich study investigating the effects of compound probiotic supplementation on the gut microbiota composition and function of captive forest musk deer, in comparison to seasonal and age-related influences. The research is timely and highly relevant to conservation physiology and animal health, offering valuable insights into microbiome-based interventions for improving the management of endangered species in captivity. The study is scientifically robust, employing high throughput 16S rRNA sequencing, a broad range of diversity and community structure analyses, and functional predictions through multiple databases. However, the confounding of treatment with geographic location, the reliance on predictive rather than empirical functional data, and the lack of longitudinal tracking limit the generalizability and mechanistic depth of the findings. Future studies incorporating metagenomics, metabolomics, and multi-site probiotic interventions are warranted. Minor revisions addressing these limitations would enhance the manuscript’s scientific rigor and conservation impact.
Response 1: Thank you very much for your positive and encouraging evaluation of our manuscript. We sincerely appreciate your recognition of the scientific rigor, methodological design, and conservation relevance of our study. Your constructive comments are truly valuable and will guide the future refinement of our research. We agree that the use of predictive functional analysis has certain limitations compared to empirical metagenomic or metabolomic approaches. While such methods were beyond the scope of the current study, we have acknowledged this in the discussion and emphasized the importance of integrating metagenomic and metabolomic tools in follow-up research to gain a deeper understanding of microbial functions. We also concur with your suggestion that longitudinal tracking of individuals and probiotic interventions across multiple sites would improve the generalizability and mechanistic insight of the findings. These directions have been explicitly stated as important future research goals in the revised version. Once again, thank you for your thoughtful and constructive feedback, which has helped us further improve the manuscript.
Comment 2: Lack of Longitudinal Data: The study is cross-sectional, capturing snapshots rather than dynamics. This limits conclusions on causality or lasting effects of probiotics.
Response 2: Thank you very much for the valuable comments. We appreciate the reviewer’s insightful comment regarding the limitations of our cross-sectional study design. We fully acknowledge that the current analysis captures only a single time point, which restricts our ability to draw definitive conclusions about temporal dynamics, causality, or the sustained effects of probiotic intervention on gut microbiota. In response, we have revised the discussion section to explicitly address this limitation and to emphasize the need for future longitudinal studies.
Comment 3: Probiotics were only administered in one location (Shaanxi): Thus, effects attributed to probiotics may also reflect environmental or management differences between farms.
Response 3: We appreciate the reviewer’s critical observation. It is indeed a limitation of our study that probiotics were administered only in one location (Shaanxi), which may introduce confounding effects due to environmental or farm management differences. We have now acknowledged this limitation in the revised Discussion section. We thank the reviewer for highlighting this important point, which will help improve the rigor of future research designs.
Comment 4: The reliance on PICRUSt-based predictions for functional profiling lacks direct evidence. Metabolomic or transcriptomic data would have strengthened conclusions.
Response 4: We appreciate the reviewer’s insightful comment regarding the limitations of PICRUSt-based functional predictions. We fully acknowledge that while PICRUSt offers useful insights into potential microbial functions, it does not provide direct evidence of metabolic activity. Incorporating metabolomic or transcriptomic data would indeed offer a more comprehensive and accurate understanding of the gut microbiota’s functional contributions. Therefore, we have revised the manuscript to clearly state this limitation and will consider integrating multi-omics approaches in future research to strengthen functional interpretations.
Comment 5: Although probiotics were the intervention, other dietary components were not strictly detailed, which may influence microbial outcomes. Provide table showing the feed composition of the other groups
Response 5: Thank you for your valuable comment. According to our communication with the farm staff, the forest musk deer in all groups had consistent access to sufficient food throughout the study, and no feed shortage occurred. The diet generally consisted of a combination of concentrate and roughage. The concentrate mainly included vegetables such as carrots, potatoes, corn, and pumpkins. The roughage was primarily composed of fresh leaves during the warm season and dried leaves in the cold season. These feeding practices were consistent across all groups except for the probiotic supplementation in the treatment group.
Comment 6: Use shotgun metagenomics or metabolomics in future work to validate predicted functional changes.
Response 6: Thank you for your insightful suggestion. We agree that incorporating shotgun metagenomics or metabolomic approaches would provide more direct and comprehensive evidence for functional changes. As suggested, we have now added a statement to the Discussion section highlighting our intention to apply metagenomics, metabolomics, or transcriptomics in future studies to validate the predicted microbial functions and improve the robustness of functional interpretation.
Comment 7: Consider a longitudinal design to monitor the lasting effects of probiotics and seasonal adaptation.
Response 7: Thank you for this valuable suggestion. We fully agree that a longitudinal study design would offer important insights into the lasting effects of probiotics and seasonal influences on the gut microbiota of forest musk deer. In response, we have revised the Discussion section to highlight this point. Specifically, we now note that future studies will incorporate longitudinal sampling to capture temporal dynamics and better assess the influence of age, seasonality, and probiotic supplementation on microbial communities and host health. We also acknowledge that applying probiotic interventions across multiple standardized sites will be necessary to control for potential environmental or management-related confounders.
Comment 8: Implement multi-site probiotic treatments to separate location from treatment effects.
Response 8: Thank you for your insightful suggestion. We acknowledge that applying the probiotic treatment at a single site in the current study may limit our ability to disentangle treatment effects from potential site-specific environmental or management factors. In future work, we plan to implement probiotic interventions across multiple standardized sites to better isolate the specific effects of probiotic supplementation from location-related variability. This approach will help improve the generalizability and robustness of the findings.
Comment 9: Clarify and standardize non-probiotic dietary components across groups.
Response 9: Thank you for highlighting this important concern. In our study, all groups received an identical basal diet, with the only variation being the supplementation of probiotics in the treatment group. Additionally, we controlled for other potential confounding variables, such as seasonal and age-related effects, during the analysis of the gut microbiota in captive forest musk deer. In future research, we will continue to rigorously standardize all non-probiotic dietary components to further minimize confounding factors and enhance the reliability of treatment-related effects.
Comment 10: Please check line 251….To identify differential abundant taxa is safe rather than species…Replace species there with taxa, and do that throughout the paper
Response 10: Thank you for your valuable suggestion. As advised, we have replaced the term “species” with “taxa” in line 251 and carefully reviewed the entire manuscript to ensure consistent and appropriate use of “taxa” when referring to differential abundance results. All relevant instances have been corrected accordingly.
Thank you very much for your affirmation of this manuscript and valuable comments again.
Reviewer 3 Report
Comments and Suggestions for Authors
Line numbers 12 to 14: please rewrite this statement. abrupt transitions and ambiguity in the wording.
Line numbers 42 to 45: Splitting the sentence makes it easier to read.
Line numbers 92 to 94: The classifications of gut bacteria are oversimplified and should be cited.
Line numbers 218 to 219: For separate tools, reference [38] is mentioned twice in a single phrase; please double-check the citations.
Line numbers 253 to 256: The sentence that lists LDA levels needs additional meaning and is far too mechanical.
line numbers 640 to 666: The assertion that the benefits of probiotics " exceeded those of season and age" has to be supported by precise effect size measurements as opposed to merely a comparison analysis.
Figure legends are very complicated and employ technical terms without providing context.
Group identification and the color scale must be simpler to understand at a glance.
Limitations:
- The first limitation is related to the composition of probiotics. There is no thorough explanation of the probiotic combination. Which concentrations and strains were used?
- Second is about seasonal bias. Only summer and winter are included in the seasonal samples; transitional dynamics may be shown in the spring and fall.
- Third is about the data related to health outcomes. No specific health measures (such as morbidity/mortality or illness scores) are provided to substantiate the study's conclusion that probiotics enhance microbial balance.
- Fourth is about geographical areas. Geographical or farm-specific effects are not statistically taken into account, even though samples were collected from two farms in separate provinces (Qinghai and Shaanxi). This might be a confounding factor.
For clarity and professionalism, the paper must be corrected for several grammatical and stylistic errors. Native speaker confirmation is required.
Author Response
Comment 1: Line numbers 12 to 14: please rewrite this statement. abrupt transitions and ambiguity in the wording.
Response 1: Thank you for your valuable comments. In response to the reviewer’s suggestion, we have revised the sentence in lines 12 to 14 to improve the clarity and ensure a smoother transition in wording.
Comment 2: Line numbers 42 to 45: Splitting the sentence makes it easier to read.
Response 2: Thank you for the valuable comments. In response to the reviewer’s suggestion, we have revised and split the sentences in line numbers 42 to 45 to improve clarity and readability.
Comment 3: Line numbers 92 to 94: The classifications of gut bacteria are oversimplified and should be cited.
Response 3: Thank you for your valuable comments. In response, we have revised the description regarding the classification of gut bacteria and added appropriate references to support the modifications.
Comment 4: Line numbers 218 to 219: For separate tools, reference [38] is mentioned twice in a single phrase; please double-check the citations.
Response 4: Thank you for the reviewer’s comment. After careful verification, we confirm that both the phylogenetic tree construction using IQ-TREE and the model selection using ModelFinder were performed as part of the same analytical workflow. Therefore, reference [38] (Kalyaanamoorthy et al., 2017) was cited twice, as it is relevant to both procedures.
Comment 5: Line numbers 253 to 256: The sentence that lists LDA levels needs additional meaning and is far too mechanical.
Response 5: Thank you for the reviewer’s valuable comments. In response, we have added an explanation following line numbers 253 to 256 regarding the significance of LDA in gut microbiota research. Specifically, LDA scores are used to quantitatively assess the enrichment levels of specific microbial taxa across different groups and their contributions to intergroup differences, with higher LDA scores indicating a more significant role of the taxa in distinguishing between groups.
Comment 6: Line numbers 640 to 666: The assertion that the benefits of probiotics " exceeded those of season and age" has to be supported by precise effect size measurements as opposed to merely a comparison analysis.
Response 6: Thank you for the reviewer’s valuable suggestions. Based on the comprehensive analyses presented in our results—specifically, the β-diversity analysis of gut microbiota composition under different influencing factors (Section 2 in Results), the comparison of dominant bacterial taxa among groups (Section 3 in Results), the impact of different factors on gut microbiota composition and diversity (Section 4 in Results), and the functional differences in gut microbiota among groups (Section 5 in Results)—we conclude that the effect of compound probiotics on the gut microbiota of captive forest musk deer is greater than that of seasonal and age factors. Furthermore, the Variation Partitioning Analysis (VPA) in Section 4 quantifies these effects, showing that compound probiotics, season, and age explain 31.09% (with 23.82% as unique explanation), 7.17% (1.75% unique), and 3.90% (1.54% unique) of the community variation, respectively, thereby confirming that compound probiotics have a more pronounced influence on gut microbiota differences compared to seasonal and age factors.
Comment 7: Figure legends are very complicated and employ technical terms without providing context.
Response 7: Thank you for the valuable comments. We agree with the reviewer’s suggestions and have thoroughly revised the figure legends by removing redundant information. The revised figure legends are now more concise and accurately convey the intended content.
Comment 8: Group identification and the color scale must be simpler to understand at a glance.
Response 8:
We appreciate the reviewer’s suggestion regarding the clarity of group identification and color scale. In our study, we aimed to investigate the effects and contributions of three factors—probiotics supplementation, season, and age—on the gut microbiota of captive forest musk deer. Therefore, six distinct groups were designed to reflect different combinations of these variables. To ensure that each group is clearly distinguished in the figures, we used distinct colors for visual separation. While we did not modify the color scheme in the current revision, we have carefully re-checked the figure legends and annotations to improve clarity and ensure that the group definitions are easy to identify at a glance. We hope that the current presentation is now more comprehensible to readers.
Comment 9: The first limitation is related to the composition of probiotics. There is no thorough explanation of the probiotic combination. Which concentrations and strains were used?
Response 9: Thank you for pointing out this important limitation. The probiotic supplement used in this study was provided as part of routine feeding management, and the information regarding its exact composition (including specific strains and concentrations) was obtained from the animal caretakers. Unfortunately, detailed formulation data were not available to us at the time of the study. We acknowledge this as a limitation and have added a statement in the discussion accordingly. In future studies, we will ensure full documentation of probiotic formulations to allow for clearer interpretation and reproducibility.
Comment 10: Second is about seasonal bias. Only summer and winter are included in the seasonal samples; transitional dynamics may be shown in the spring and fall.
Response 10: Thank you for your thoughtful comment. In this study, we selected winter and summer as representative seasons, as they typically exhibit the most distinct environmental and physiological conditions for captive forest musk deer. Moreover, our previous published work has already investigated the seasonal dynamics of gut microbiota across all four seasons, including spring and fall (Jiang et al., 2020). Based on those findings, we focused on the two most contrasting seasons in this study to better capture the seasonal divergence in gut microbial communities. We have now clarified this rationale in the revised manuscript. We agree that including transitional seasons would provide further insights, and we will consider incorporating them in future studies to explore microbial dynamics with higher temporal resolution.
Comment 11: Third is about the data related to health outcomes. No specific health measures (such as morbidity/mortality or illness scores) are provided to substantiate the study's conclusion that probiotics enhance microbial balance.
Response 11: Thank you for your valuable suggestion. In this study, the information on probiotic use was obtained from the animal keepers, and the main focus was to evaluate the relative contributions of probiotic supplementation, season, and age to the gut microbiota composition of captive forest musk deer, rather than to assess the direct effects of probiotics on individual health outcomes. Therefore, detailed health metrics such as morbidity, mortality, or clinical illness scores were not recorded. We agree that incorporating such data would greatly strengthen the interpretation of probiotic effects, and we will consider including these health-related indicators in future research to better understand the relationship between gut microbiota modulation and animal health.
Comment 12: Fourth is about geographical areas. Geographical or farm-specific effects are not statistically taken into account, even though samples were collected from two farms in separate provinces (Qinghai and Shaanxi). This might be a confounding factor.
Response 12: Thank you for your thoughtful comment. We acknowledge that geographic or farm-specific factors may act as potential confounders. Although the two farms are located in different provinces (Qinghai and Shaanxi), our observations indicated that the composition of the feed provided to the forest musk deer was quite similar between the two farms. In this study, we focused on evaluating the effects of probiotic supplementation under controlled conditions of age and season. However, we did not statistically control for geographical variation. We agree that this is an important limitation and will consider addressing it in future studies by conducting experiments within the same farm to minimize potential geographic and environmental confounding effects.
Comment 13: For clarity and professionalism, the paper must be corrected for several grammatical and stylistic errors. Native speaker confirmation is required.
Response 13: Thank you for your helpful suggestion. In response to your comment, we have carefully revised the manuscript to improve its language quality, with particular attention to grammar, scientific terminology, and sentence clarity. The revised version has also been reviewed by a native English speaker with a background in scientific writing to ensure clarity and professionalism.
Thank you very much for your affirmation of this manuscript and valuable comments again.
Round 2
Reviewer 3 Report
Comments and Suggestions for Authors
There is no information on health outcomes? Conclusions about "improved health" are yet unclear and have not been quantified.
To further lessen any claims regarding health and make sure the location of the farm confounder is mentioned explicitly in the Results discussion and Conclusion, I suggest making a few minor revisions before publication.
The paper must be corrected for some grammatical and stylistic errors.
Author Response
Response 1: Thank you for your valuable comment. We fully agree that direct health outcomes were not measured in this study, and therefore we have revised the conclusion to avoid making unsubstantiated claims regarding improved health. The revised conclusion now emphasizes our findings on gut microbiota modulation and positions them as a basis for future studies aimed at quantifying the potential health benefits of probiotic supplementation in captive forest musk deer.
Comment 2: To further lessen any claims regarding health and make sure the location of the farm confounder is mentioned explicitly in the Results discussion and Conclusion, I suggest making a few minor revisions before publication.
Response 2: Thank you for this valuable suggestion. In response, we have revised the Discussion to further temper statements regarding health benefits, clarifying that our findings should be interpreted as patterns in gut microbiota modulation rather than direct evidence of improved health. We have also explicitly acknowledged the potential confounding effect of farm location, noting that the probiotic group was limited to a single site and that environmental or management factors may have influenced the observed microbial differences. This limitation has now been discussed in detail, and we have indicated that future studies will implement probiotic treatments across multiple standardized sites to better separate location effects from treatment effects. The revised text can be found in the Discussion section.
Comment 3: The paper must be corrected for some grammatical and stylistic errors.
Response 3: Thank you for your valuable suggestion. In response, we have thoroughly revised the manuscript to enhance its linguistic quality, focusing on grammar, scientific terminology, and sentence conciseness. Additionally, the revised version has been reviewed by a native English speaker with expertise in scientific writing to ensure clarity, accuracy, and professionalism.
Thank you very much for your affirmation of this manuscript and valuable comments again.